# A Blockchain-Based Authentication Mechanism for Enhanced Security

**DOI:** 10.3390/s24175830

**Published:** 2024-09-08

**Authors:** Charlotte McCabe, Althaff Irfan Cader Mohideen, Raman Singh

**Affiliations:** School of Computing, Engineering and Physical Sciences, University of the West of Scotland, Lanarkshire G72 0LH, UK; charlotte.mccabe@alumni.uws.ac.uk (C.M.); althaff.mohideen@uws.ac.uk (A.I.C.M.)

**Keywords:** authentication, blockchain technology, smart contracts, multi-factor authentication, cryptocurrency

## Abstract

Passwords are the first line of defence against preventing unauthorised access to systems and potential leakage of sensitive data. However, the traditional reliance on username and password combinations is not enough protection and has prompted the implementation of technologies such as two-factor authentication (2FA). While 2FA enhances security by adding a layer of verification, these techniques are not impervious to threats. Even with the implementation of 2FA, the relentless efforts of cybercriminals present formidable obstacles in securing digital spaces. The objective of this work is to implement blockchain technology as a form of 2FA. The findings of this work suggest that blockchain-based 2FA methods could strengthen digital security compared to conventional 2FA methods.

## 1. Introduction

To safeguard digital spaces from cyber threats, authentication plays a vital role in ensuring sensitive information remains inaccessible to unauthorised parties. However, modern two-factor authentication (2FA) methods may suffer from security risks. This can include verification code forwarding attacks (VCFAs), low-entropy one-time passwords (OTPs), exposure of biometric data, and more. A survey done by the Department for Science, Innovation, and Technology (2023) found that 32% of businesses and 24% of charities in the UK recognised a breach or attack in winter 2022–2023 [1].

In response to these challenges, this research proposes a protocol for an authentication method to potentially replace one-time passwords (OTPs) and other forms of 2FA. For demonstration purposes, this is implemented using a web application. The Ethereum blockchain is utilised to implement a 2FA method in which the user will authorise interactions within the web application and add these as transactions to the blockchain. The different phases, such as registration, login, password change, and account deletion, are demonstrated. This will be used instead of asking the user to request and wait for an OTP.

The research will also investigate various attack vectors targeting authentication, detail the issues with using databases (e.g., MySQL, MongoDB) for authentication, and provide an explanation as to why new techniques need to be developed. This study aims to contribute to the advancement of authentication technology and fortify it against evolving cyber threats.

In conclusion, this research presents a different and uncommon approach to the implementation of 2FA by integrating the Ethereum blockchain to the authentication process. Blockchain technology applied to authentication protocols is relatively untapped but is a promising method for enhancing security.

## 2. Background

The origins of distributed databases, of which blockchain is a type, go back as far as the 1970s. However, the 2008 seminal paper by Satoshi Nakamoto, the Bitcoin creator, is where the popularity of blockchain technology was set in motion [2]. While the Bitcoin creator did not use the term “blockchain” in their paper, the term was given to the design behind the operations of the Bitcoin currency [3].

Blockchains are a way in which mutually distrustful parties (nodes) can come to an agreement regarding the condition of a ledger that records information. A blockchain is a distributed ledger and contains records (blocks) of information regarding transactions between multiple parties. These blocks are connected cryptographically, transforming into an immutable ledger [2]. The starting block in the blockchain is called the genesis block. As every block also stores the hash of the previous block, tampering with a block will result in its hash changing. The hash of a block is generated by a combination of all data within that block. Any changes to this will make subsequent blocks invalid. On top of this, if an attacker attempts to make a tampered block valid, they will also have to redo the proof of work (PoW). PoW is a technique used to verify new transactions on the blockchain for the current block and all succeeding blocks. This requires time and much computational power, and it is highly unlikely to succeed, as the attacker would need to take control of over 50% of the blockchain network without being noticed [4]. As PoW requires its validators to solve a puzzle in order to update the blockchain and receive their compensation, the probability of solving the puzzle first depends on raw computational power. Validators are competing through their massive energy expenditure. Due to this, it has been argued that Bitcoin consumes electricity at a rate comparable to Austria or Ireland. To reduce the exorbitant amount of energy used, several alternatives to PoW have been suggested, the most notable being proof of stake (PoS). PoS instead randomly selects a validator to solve the puzzle and, in turn, update the blockchain. This takes away the need for competition between validators and reduces overall energy consumption. However, unlike PoW, PoS does not require its validators to incur a monetary cost once gaining the authority to update the blockchain. This means a validator will always update the ledger, even if the update causes disagreements. Some claim for this reason that PoS fails to generate consensus [5].

Information can be appended to the blockchain by nodes through transactions. The blockchain has policies that regulate what is allowed to happen to the blockchain. These include the following. The access policy dictates who is allowed to read the information. The control policy dictates which nodes are allowed to participate in the development of the blockchain and how new blocks should be appended. A range of control policies such as private, permissioned, permissionless, and consortium are available. Finally, the consensus policy dictates what is valid within the blockchain and resolves conflicts. Additionally, some blockchains also support the deployment of smart contracts. These execute the terms that the parties have agreed upon [2].

### 2.1. Blockchain Applications

Blockchain can be potentially used in three main fields. The first and most common is the use of digital payments. Contemporary commercial structures for payment authorisations rely on centralised ledgers. These ledgers record all transactions and account balances. Simply, the transactions are sent once from the party involved in the transaction. Then, the transaction is sent to an intermediary, where it is checked for validity before the accounts are updated. A blockchain transaction, such as Bitcoin, can be a slow and expensive process, as the transaction is sent to every member node, where more transmissions will take place. Eventually, this transaction is sent to every node computer after it is appended to the blockchain. This involves more time and processing power. For this reason, vendors will use a centralised clearance, as they can clear more transactions per second compared to a decentralised cryptocurrency like Bitcoin. However, blockchain is favoured by companies like Bitcoin, as it eliminates the need for trusted third-party intermediaries. As every member node competes to verify transactions, the need for ‘trusted’ nodes is removed [3].

The second way in which blockchains can be used is in the deployment of smart contracts. A smart contract is a code script that interacts with the blockchain to represent an agreement(s) between the parties involved without reliance on a trusted third-party intermediary. The smart contract can trigger transactions within the blockchain to execute the terms of the agreements. Once this smart contract is deployed, it cannot be altered unless specified in its design. There are three types of smart contract transactions, which include the following.

Direct Transfer: This is a monetary transfer between accounts. These transactions do not need to interact with a smart contract.

Contract Creation: This transaction is used for implementing new smart contracts that will attach its bytecode to the blockchain ledger. A bytecode is a 160-bit address that is generated for the smart contract. This is then assigned to the smart contract in the blockchain as its identifier.

Contract Execution: When an account triggers a smart contract, this transaction is generated. This is done when the account assigns the smart contract address and calls its functions to allow it to send the relevant data being asked of it by the smart contract’s parameters. When a block with a transaction is appended to the blockchain on the node, the smart contract may also trigger an event to notify that this has happened.

However, due to the limited resources for smart contract developers, it is harder for developers to find and implement best practices for security within their smart contracts [6]. This was discovered when the first smart contracts were implemented on the Ethereum network called the Decentralized Autonomous Organization (DAO). An attacker was able to execute the code of the smart contract to send a third of the USD 150 million invested in this smart contract to themselves. Due to this attack, the Ethereum blockchain was rolled back to reverse this but created a split in the network. Two networks now exist: the one with the DAO attack and the other with it reversed. Due to this fork, the claim of immutability of the Ethereum network has been questioned [3].

A third way that blockchains can be utilised is in databases and record management. Records must remain trustworthy, especially when they are critical to an underlying infrastructure such as land registries, births/deaths/marriages, and financial transactions. It would be catastrophic if the integrity of these records were compromised. For example, counterfeit civil registrations could be an opportunity for immigration fraud and endangering national security [7]. Blockchain characteristics such as immutability and process integrity are appealing qualities, especially in record management. However, the data capacity constraints are expensive, and delivery performance can be controversial [8].

The final possible application of blockchain technology that is discussed in this research is that of authentication of user data. In order to understand why this could be useful, an examination of vulnerabilities in software and technologies associated with authentication must be discussed.

### 2.2. Authentication Methods

Authenticating users by a username and password combination is routinely used in online services. Typically, a client will first register for their account with the service by creating a username and password pair. They will then be required to show they know the pair before being allowed access to their online account. Other than the user remembering the pair, the authentication process usually does not require any other authentication. Human memorable passwords are vulnerable to guessing attacks due to being predictable. An attacker can repeatedly send authentication requests until a correct username/password combination is found. This type of attack can be thwarted by implementing an account lock once a certain number of attempts to enter the account has been reached. However, this method can increase customer service costs for legitimate users who have locked themselves out of their accounts. It can also increase the risk of denial of service (DOS) attacks and downtime for the service. In the case of offline guessing attacks, no interaction with an authentication server is required, and passwords can be found easily by just observing a successful authentication session between a client and server. Offline guessing attacks can be mitigated using symmetric and asymmetric methods. Symmetric methods require the authentication server to have password-derived data, e.g., a password table of all clients. Asymmetric methods require the server to have both password-derived data and the private key of its public key [9].

Symmetric algorithms use the same cryptographic keys for both encryption and decryption. Asymmetric algorithms use a pair of keys. Public keys are publicly available, and private keys are only known to the owner. The public key is used for the encryption, and the private key, which only the owner knows, is used for the decryption [10]. An example of this would be an online service that has a public key certified by an authority and that uses the server’s public/private key to establish a secure connection. Assuming the cryptographic methods being used are secure, eavesdropping does not pose a threat. However, despite this, passwords still need to be secure. Passwords need to be easy to remember for the owner but hard to guess for the attackers. This task is very difficult to achieve, especially when users have multiple online accounts. Also, it is recommended that users have different passwords for each account in case one is compromised. For this reason, it may be nearly impossible for a human to successfully create complex passwords for multiple accounts while remembering them all. There are multiple ways in which users can retrieve passwords, such as online password managers and web browsers that store passwords. However, the user will not remember the password, as it is automatically being entered on their behalf. Similarly, these methods also have their own inherent security risks [9].

### 2.3. Two-Factor Authentication

Two-factor authentication (2FA) is the process of reinforcing a password-based authentication process by using a secondary method such as an authentication token. It aims to provide a greater level of protection by extending single-factor authentication. It utilises what the user already knows (e.g., the password) along with what the user has (e.g., a hardware-based token) or what the user is (e.g., biometrics). This means that if one authentication method is compromised, it is more difficult for an attacker to successfully authenticate themselves unless they have access to the other authentication method. One-time passwords (OTPs) are an alternative to biometric and hardware-based token authentication. Biometrics are expensive and raise privacy concerns, while hardware-based tokens are inconvenient, as a user needs to have multiple hardware tokens for each organisation. 2FA schemes utilising mobile phones are a popular 2FA method, as they provide a compromise between convenience, security, and expense.

One example of 2FA is Short Message Service (SMS)-based authentication. Its purpose is to send an OTP via SMS to a user’s phone. The user will then enter this OTP into a website or application to authenticate their login. Often, banks will use a transaction authentication number (TAN) as an OTP. TANs are cryptographically bound to the specific transaction data that is being authenticated and are only valid for this transaction. Another example of 2FA is requesting an OTP from an authenticator application on a smartphone. SMS messages are not required in this method [11].

The first security flaw of 2FA for both methods of OTP acquisition is that it only confirms that the user has access to the OTP and not the identity of the user. Anyone with access to the device, whether physically or remotely, can send themselves the OTP. SMS-based 2FA is susceptible to remote phishing such as verification code forwarding attacks (VCFAs). This attack is executed when an attacker requests a password reset from the service provider on behalf of the user. They will then proceed to send a fraudulent SMS to the user pretending to be the service provider requesting the OTP for ‘security purposes’. While this may seem an obvious phishing attack for some, a research found this technique to be very effective [12]. The study used 300 participants who were not informed that the study involved SMS phishing. In one instance, as many as 50% of participants fell prey to an SMS attack. A successor to the SMS-based 2FA is an authenticator application such as Google Authenticator. While these codes can still be requested by the user, it is potentially harder for an attacker to convince a user to give them something they already have (the code generated by the authenticator) compared to something they have been given (the code sent to them via SMS) [13].

While many organisations are moving away from SMS to authenticator-generated OTPs, there is still cause for concern. Another possible security flaw to OTP-based 2FA is the strength of the OTP itself. A low-entropy OTP is vulnerable to brute force (guessing) attacks. For this reason, the OTP must pass a standard randomness test. Also, if a service provider does not invalidate an OTP relatively quickly or if they repeat an OTP—even if the IP address, browser, or operating system has changed—then this allows an attacker to capture an OTP. They will then prevent the user from submitting it and proceed to reuse the OTP in a different login session, as it is still valid. Additionally, 2FA is vulnerable to malware deactivating it on behalf of a user. Once a legitimate user has successfully logged into their account via authenticator-assisted 2FA, malware could deactivate 2FA on the account if the service provider does not ask for additional 2FA before executing this action. As part of this research, this was tested on a Google account. As of February 2024, 2FA can be simply turned off without any additional authentication methods. As this type of 2FA requires a user to have their mobile phone on them to complete authentication, this can affect usability if the device is lost or unavailable. To combat this, service providers allow for recovery methods such as recovery passwords. This can be backup OTPs or a human-readable string. If there are no additional authentication methods before accessing these recovery codes, malware can wait until the user has successfully signed in and accessed these codes [11]. As of February 2024, this is still the case for Google accounts.

Biometric 2FA is also susceptible to exploits. While biometrics are unique to a person (e.g., fingerprints) and confirm the identity of a user compared to someone just having access to an OTP, if the database storing this biometric data is compromised, then it is rendered useless. The DNA footprint of a person cannot be changed like a password or OTP [14].

### 2.4. Password Encryption and Salting

Passwords are deemed the first line of defence against attackers who are trying to gain unauthorised access to accounts or services. For this reason, it is essential that passwords are heavily protected. While having a long and strong password is important to prevent guessing attacks, if the password is not stored securely, an attacker can find it by other means. Hashing algorithms are one of the best methods of securing a password. Cryptographic hashing algorithms are mathematical procedures used to convert data into a fixed-size hash. Hashing algorithms are designed to be non-reversible so that, even if an attacker manages to find it, they cannot convert it back to its original state. When passwords are converted and stored as a hash value, this makes it tedious for attackers to guess [15].

However, hashing algorithms can have vulnerabilities that may be exploited using brute force or dictionary attacks. In addition, hash disclosure using rainbow tables (tables with values of known hashes for a particular cryptosystem) may enable an attacker to reverse-engineer the hash if they pinpoint the encryption method used. This can happen due to leaks, password phishing, weak database security, or the use of dated hashing algorithms such as MD5 or SHA1. By examining the rainbow table, a list of commonly used passwords can be employed to find coincidences and discover the decrypted password. If an attacker breaches a system and gains access to the password hash table, they can utilise their rainbow table to successfully decrypt the passwords. It is common practice to have a password hash table to avoid loss of user data. Organisations that have weak password protection and system security are especially vulnerable to this. In addition to increasing the level of security of their system, it is recommended that organisations also use modern salting techniques. This is the principle of adding additional random values to hashed passwords to create new hash values. A system that utilises salting methods reduces the risk of successful rainbow table attacks. Modern and secure password authentication systems will use these methods [16].

### 2.5. PBKDF2

Whereas MD5, SHA1, and SHA256 hashing algorithms are vulnerable to dictionary, rainbow table, and brute force attacks, the Password-Based Key Derivation Function (PBKDF2) is designed to be computationally intensive and take longer to compute. Hashing algorithms are used to map the variable length to a fixed output, and the data are retrieved from a database. Cryptographic hash functions involve hash-based message authentication codes (HMACs) to ensure the integrity of the data by using cryptographic keys. Collision-free hashing functions are characterised by inputs never yielding identical hash values. The user’s passwords should never be used directly as a cryptographic key, as they do not satisfy entropy and randomness requirements. Strong cryptographic key generation is vital for deterring criminals from cracking them. Pseudorandom function (PRF) is a PBKDF2 method that utilises a fixed number of iterations. It accepts input in the form of a salt, the user’s password, and the desired length of the output key. It repeats the process to the number of the iteration count to produce the key. This increased computation power makes the process complicated and helps resist brute force and dictionary attacks [17]. However, PBKDF2 has also been known to have vulnerabilities [18]. Strong hashing algorithms used today include, for example, Argon 2 [19].

### 2.6. Ethereum Blockchain and Solidity

Bitcoin is one of the most popular cryptocurrencies and a public blockchain. Close behind is Ethereum, an open-source platform and public blockchain that allows decentralised applications to be deployed and run. Its fundamental innovation, the Ethereum Virtual Machine (EVM), is capable of running programmes in any language if enough time and memory are allocated. EVM is a runtime environment for smart contracts, and every node in the Ethereum network utilises it. The EVM is separate from the main Ethereum network. The benefits of Ethereum are its scalability and lower transaction (gas) fees in comparison to Bitcoin [4].

Smart contracts allow for complex requests in comparison to Bitcoin operations. Anything of value, such as money, property, and information, can be exchanged with smart contracts using a high-level language such as Solidity. To assist developers in creating smart contracts for the Ethereum blockchain, the programming language Solidity was designed to be like the JavaScript language [8].

Smart contract instances on the Ethereum blockchain are identified by a unique address, hold an amount of Ether (Ethereum’s virtual currency), and have a persistent place on the blockchain where their state is stored and associated with their executable code, which is considered immutable [20]. The blockchain also hosts externally owned accounts (EOAs), which are also identified by their unique address and hold an amount of Ether but do not have associated code. They start programmes by issuing a transaction to deploy a new contract instance or by sending messages to a contract to invoke a function. These transactions have data such as the sender address of the EOA, an amount of virtual money as payment for the contract, and a gas fee for the miner node that executes it as a reward [21]. The main Ethereum network is called the ‘Main Net’, and operations switched to PoS consensus after September 2022 [20].

Solidity is a commonly used language for smart contracts on the Ethereum blockchain. When used as object methods, state variables and functions within smart contracts can refer to the current contract instance through the variable ‘this’. Functions can also send messages to other contracts using the variable ‘msg’; this could include gas fees or monetary amounts. Additionally, ‘msg’ stores information such as the address of the sender (msg.sender) or the monetary amount sent (msg.value) [21].

### 2.7. Cryptocurrency Wallets, Test Networks, and Development Environments

For the smart contract to communicate with users, a user interface capable of interacting with the blockchain must be used. MetaMask (https://metamask.io, last accessed on 25 April 2024) is a popular cryptocurrency wallet that is used to interact with the Ethereum blockchain. It allows users to connect their cryptocurrency wallets to websites that require entry to the Ethereum network by connecting them to a node [8]. In addition, MetaMask allows users to create accounts for use in Ethereum networks while maintaining their private keys. Export and import of private keys are also allowed in MetaMask, along with the option of switching between different Ethereum networks to reflect the balance of the account of that specific network. MetaMask allows for the transfer of Ether between accounts, the ability to hold tokens, and the option to view transaction details within a blockchain explorer such as Etherscan. MetaMask achieves this by connecting to the Infura server (www.infura.io, last accessed on 25 April 2024), which manages Ethereum nodes connecting to Ethereum networks and allows for private keys to be stored locally on the user’s computer by MetaMask. This is due to the security risks of storing private keys in a third-party cloud service. MetaMask also allows for interaction with the Ethereum blockchain by using a JavaScript library (version 2023) called web3.js [22]. Other wallets, such as Mist and MyCrypto, are also popular in the Ethereum ecosystem [23].

The Web3 library is used to connect to the Ethereum network from an application. The application binary interface (ABI) is given to the Web3 library to allow it access to the deployed contract on the network. For the Web3 instance to communicate with the Ethereum network, a communication layer is required. This is also known as a ‘provider’. The provider is an intermediary between Web3 and the Ethereum network [4].

A test network (Test Net) is a means of deploying DApps for testing. These networks closely replicate the main blockchain networks, such as Ethereum Main Net, and adopt identical consensus mechanics. Sepolia is modelled after the main Ethereum network structure and uses the same consensus mechanism [24]. Ganache is a local Ethereum Test Net and is used to create a local blockchain [4].

React, created by a software engineer at Facebook, is an open-source library in which its components act as a viewing layer for JavaScript applications. React is known to be used by Facebook (www.facebook.com, last accessed on 10 June 2024), Instagram (www.instagram.com, last accessed on 10 June 2024), and Netflix (www.netflix.com, last accessed on 10 June 2024) [25].

Truffle Suite is an application used for the development of DApps [26]. Truffle allows for a user’s computer to be set up as a local Ethereum virtual machine for the development and testing of smart contracts [27].

## 3. The Proposed Authentication Mechanism

The inherent security flaws of 2FA, along with databases such as MySQL/MongoDB used for the storage of user credentials, are a cause for concern. For this reason, this research proposes an authentication method using the Ethereum blockchain. This negates the need for 2FA mechanisms such as OTPs.

### 3.1. Design

The research involves creating a web application using various components to authenticate user activity using the blockchain instead of commonly used databases such as MySQL or MongoDB. A cryptocurrency wallet is used to authenticate the user when creating and logging into an account along with other interactions. This is done by obtaining the wallet address used during the account creation to ensure only the individual with that wallet address can access their account. The development and testing platform Truffle is used to assist in the development of the smart contract, and ReactJS is used for the front-end of the web application. Truffle is a popular framework due to its inbuilt features for testing smart contracts and was chosen for this project for these reasons [28]. React was chosen for this project due to its scalability, maintainability, and ease of development of applications and user interfaces for developers of varying skill levels [29].

Node.js and its modules are needed to create the application and deploy the contract to the Ethereum network [4]. The Truffle-generated file ‘Truffle-config.js’ is configured to host the application at the address ‘127.0.0.1’. This address is the local host network, and only devices on the local network are able to access the web application. For this project, the web application has not been made public. This is to ensure no one else initiates transactions while testing performance parameters. Passwords are salted and encrypted using the hashing algorithm PBKDF2. As discussed previously, PBKDF2 has been known to have vulnerabilities. Alternative algorithms such as Argon 2, as discussed, are stronger; however, due to technical difficulties with implementation in React and insufficient time to troubleshoot, PBKDF2 is used instead for testing purposes.

The front-end of the application was created using the JavaScript language. This includes three web pages, including the registration, login, and home pages. Other files were created using JavaScript, including the App.js file. This file handles the routing logic for the web application. Routing in ReactJS can be accomplished using the react-router-dom library. This informs the application of the paths to navigate to for each web page. Another JavaScript file, web3helpers.js, was created to house Web3 functions responsible for fetching addresses from MetaMask.

For this project, MetaMask is the cryptocurrency wallet of choice. This is because it is browser-based, works with Chrome or Firefox, and is relatively quick to set up. To read data from a transaction or a block in Ethereum, a block explorer and analytics platform such as Etherscan is required (accessible at www.etherscan.io). This application allows users to view the details of a block, such as confirmation time, hash, transaction price, gas price, and more [23].

The local blockchain environment was established using the programme Ganache. This aids in the development and testing of the web application. The project was then deployed to the Ethereum Test Net Sepolia for full-blown implementation. Ganache was configured to the network ID and the remote procedure call (RPC) server. MetaMask was configured to use these credentials. The backend of the application was built with Truffle. The smart contract was created using the programming language Solidity. A deployment JavaScript file was created to instruct the application to deploy the smart contract. The smart contract handles account registration and listens for successful login attempts, password changes, and account deletion. It emits all these events to the blockchain.

Truffle is also used to create an address for the smart contract. Ganache generates private keys to allow for import into MetaMask. These wallets are awarded ETH 100 Ganache coins to allow for the deployment and testing of the smart contract. Each transaction (interaction from the user) requires a certain amount of gwei (Ethereum sub-currency) to execute. Once the web application has finished its development and testing in Ganache, it is finally deployed to Sepolia. This helps analyse the performance of the blockchain that would not be possible on a local Test Net. The application requires some changes to the file ‘Truffle-config.js’ in order to be deployed to the Sepolia Network. To access the Sepolia Test Net, an account with ‘www.infura.io’ needs to be created. Infura then provides an API key for the project to allow it to connect to its Ethereum node.

To start deploying on Sepolia, Test Net Ether is required. An organisation such as Alchemy allows developers to request ETH 0.5/day to help test and troubleshoot decentralised applications. Once ETH 0.5 has been sent to the MetaMask wallet, testing of the application can begin. For this project, the smart contract is not deployed to the Ethereum Main Net. This is due to monetary restraints, as Test Net coins are not allowed on the Ethereum Main Net. Once the application is running, MetaMask needs to connect to the application before the user can use it. Once deployed, transactions the wallet is making to the blockchain can be viewed using the EtherScan utility (www.sepolia.etherscan.io, last accessed on 15 July 2024).

The following performance parameters are analysed to investigate how practical the application is for intended use in a real blockchain environment and how certain parameters could affect user experience.

Transaction Throughput: The number of transactions processed per second.

Response Time: The amount of time taken for the transaction to be initiated and added to the blockchain.

Latency: The time delay between the user clicking the button to call MetaMask and confirmation from the application that it has acknowledged a successful transaction from MetaMask.

Comparison: The delay yielded from the latency test is compared to the login page of another service provider with 2FA enabled. This helps understand if this authentication method is slower or faster than 2FA.

Scalability: The ability of the network to handle an increasing number of transactions without sacrificing performance such as latency and throughput.

Transaction Fees: How much gwei is spent per transaction.

Samples from each parameter will be taken at busy times and quiet times on the Sepolia network. An average will be calculated for each parameter to obtain an accurate insight into how the application performs in a real environment.

### 3.2. Sample Size

The size of samples taken for this research is relatively small due to time constraints. As discussed by [30] regarding sample sizes using tornado occurrence data, these can significantly impact conclusions that can be drawn from the data being analysed. While larger sample sizes generally allow for better hypothesis tests and can allow for smaller effects to be detected, having a large dataset may also not guarantee adequacy, depending on sample variability. Small samples are likely to be sufficient for concluding if the variability is low. Samples with high variability are less consistent and make conclusions and predictions hard to determine. Due to the constantly changing gas prices of the Ethereum network and the unpredictability of cryptocurrencies in general, variability is high. While this research would have benefited from a larger sample set, conclusions could still be difficult to determine due to the nature of cryptocurrency.

## 4. Implementation and Deployment

The application takes the form of a simplistic user login page with the option to create a new account. Upon creation of an account and the subsequent login, these are recorded as transactions to the blockchain. The home page allows the user to either change their password or delete their account. These actions are also recorded on the blockchain. The user also has the option to log out of their account; however, this aspect is not discussed in the present research. Figure 1 shows the flowchart of the proposed mechanism.

### 4.1. React

A new React project was created using the following command: *npx create-react-app blockapp*. Various libraries were installed for the application to work using the Node.js package manager (npm). These libraries include the following:@Truffle/hdwallet-provider: This library establishes a network for connecting to Ethereum through Infura. It also provides Truffle with required functionalities that are not offered by Infura. This includes event filtering and transaction signing [31]. This library is required to connect to Sepolia through Infura.crypto-js: This is a JavaScript library that has a collection of cryptographic algorithms, such as Advanced Encryption Standard (AES) [32], Public Key Cryptography Standards (PKCS) [33], Password-Based Key Derivation Function (PBKDF2) [18], and more [34]. This library is used for password encryption using PBKDF2.react: This library is used to develop user interfaces. It allows for the development of large, complex web applications without the need for subsequent page refreshes after data are changed [35]. This library is installed when using the command npx create-react-app.react-dom: A core library of React. It is a renderer for browser environments or server-side rendering [36].react-router-dom: Used for defining paths and its associated components [37].react-scripts: Includes scripts and configurations used by the React app [38].web-vitals: Enables the reporting of web vitals to aid in performance improvement.web3: Allows the application to interact with a local or remote Ethereum node.solc-js: The node module ‘solc’ is required for compiling the Solidity file. This gives the Solidity code its bytecode and ABI [4].axios: A promise-based HTTP client for node.js [39].

After the Register.js, Login.js, and Home.js pages are created and placed in the correct directories within the React project, Truffle is used to build the project using the command truffle build.

Once Truffle has built the project, it creates three folders called migrations, contracts, and build. A Migrations.sol contract is automatically generated by Truffle for managing migrations between different versions of smart contracts. A deploy_migration.js file is created to instruct it to deploy the Authenticate.sol contract. Using the command truffle migrate-reset instructs Truffle to create an address for both contracts. Ganache is configured to the network ID ‘1337’, and the remote procedure call (RPC) server to ‘HTTP://127.0.0.1:7545’. From Ganache, a private key is copied and pasted for import into MetaMask. Once the account is imported into MetaMask, a new network is configured within the settings to use Ganache credentials. MetaMask needs to be switched to the Ganache network to connect to the web application. Once MetaMask and Ganache are configured successfully, the web application can be started by using the command npm run start.

Upon successful startup, MetaMask asks to connect the wallet to the application. From here, the full functionality of the web application can be utilised and tested before deployment to the Sepolia network. To connect to Sepolia, the truffle-config.js file needs to be configured to the network ID ‘11155111’, the Infura project ID URL, along with the mnemonic phrase for the MetaMask wallet. Truffle is used again to create contract addresses on the Sepolia network using the command truffle migrate–reset–network sepolia. Upon starting the web application, all transactions are now recorded to the Sepolia blockchain. This can be viewed using the blockchain explorer Etherscan.

### 4.2. Main Code

A smart contract called Authenticate is created using the Solidity programming language. The contract hosts a struct named ‘UserAccount’ to store the data types for the user’s email, salt, password, and wallet address. A public mapping is created called ‘users’ to allow for strings to be used as keys. Each key is associated with the values in the ‘UserAccount’ struct. Events are defined for each stage of user interaction with the application, such as the creation of a new account, successful login to account, account password change, and account deletion. These events log and notify the blockchain regarding occurrences within the smart contract. The smart contract has four functions to handle user interaction within the application and emit the events to the blockchain. These are called: ‘CreateAccount’, ‘LoginAttempt’, ‘DeleteAccount’, and ‘UpdatePassword’. For testing purposes, the ‘NewAccountCreated’ and ‘PasswordChanged’ events emit full credentials, such as salt and hashed password, using dummy user credentials. In a real setting, exposing this information would be detrimental to account security. Listing 1 shows the Solidity structure schema for the user account.

**Listing 1.** Solidity implementation of struct, public mapping, and events.

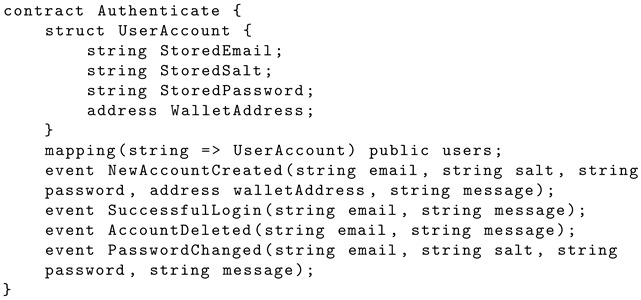



### 4.3. Create Account

The ‘CreateAccount’ function assigns a new user account to the ‘users’ mapping and emits this event to the blockchain. The email, salt, hashed password, wallet address, and string message are emitted. Listing 2 shows the Solidity structure schema for account creation functionality.

**Listing 2.** Solidity implementation of ‘CreateAccount’ function.

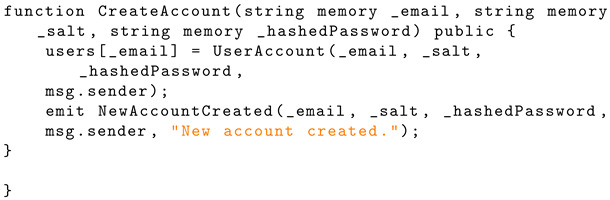



The registration page imports the following libraries:useState: The hook ‘useState’ returns an array with the value of the current state and the set function [40].useEffect: This hook allows for synchronisation of a component with external systems [41].useNavigate: This hook imported from ‘react-router-dom’ allows for navigation to a URL based on a path. In this project, the application navigates to either ‘localhost:3000/Login’, ‘localhost:3000/Register’, or ‘localhost:3000/Home’ [42].loadWeb3: A function created within the web3helpers.js file to establish Web3 for interaction with Ethereum.loadBlockchainData: A function created in the web3helper.js file to load data such as the Ethereum account, network ID, and contract instance. Listing 3 shows the Solidity structure schema for the registration page.

**Listing 3.** Imports used in JavaScript implementation of the registration page.

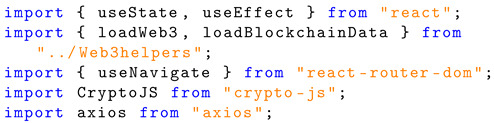



The state variables within the functional component are declared with the hook ‘useState’. An array with two elements is returned: the current state value and a function that will update the state. The ‘useNavigate’ hook is invoked with the name ‘nav’. Listing 4 shows the Solidity structure schema for React hook functionality.

**Listing 4.** React hook implementation.

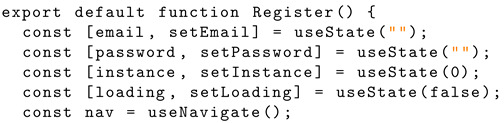



Listing 5 shows the Solidity structure schema creating a new instance for user accounts from blockchain. This function returns the property ‘auth’ (the contract instance) from the blockchain using ‘loadBlockchainData’ and a catch error.

**Listing 5.** Implementation of variable ‘userAccounts’.

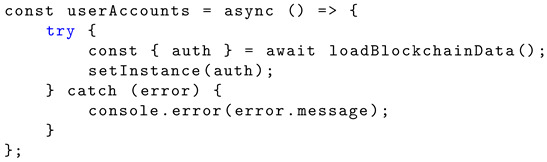



Listing 6 shows the Solidity structure schema for user password validation functionality. This code checks that an email and password have been entered by the user. There is also a requirement for a valid email address format. This means including an ‘@’ symbol and a full-point character, as is typical for an email address. A password requirement is enforced to ensure the user creates a password that satisfies the following requirements. This includes having a password length of eight characters, one lowercase character, one uppercase character, one numerical character, and one special character. Using the website ‘www.worldtimeapi.org’ (accessed on 13 May 2024), a timestamp is logged for testing.

**Listing 6.** Implementation of ‘passRequirements’ and ‘registerAccount’.

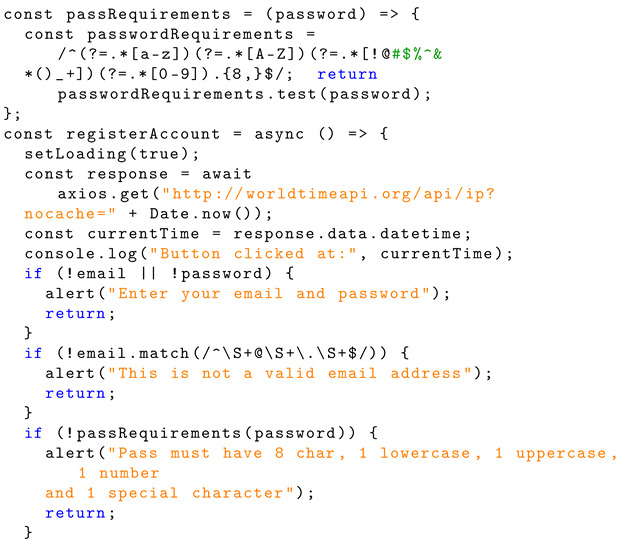



Listing 7 shows the Solidity structure schema for password salting and hashing. Using ‘window.ethereum’ and method ‘eth_requestAccounts’, a request is sent for the user’s Ethereum account. If an Ethereum account is not found, it alerts the user. The wallet address for the user is assigned to the variable ‘walletAddress’. A salt is generated using the JavaScript library CryptoJS. The length of the random value that is generated is specified as 128 bits divided by 8 bits, to make a 16-byte key. The hash of the password is generated using the salt that was generated previously and uses the library CryptoJS again to call the hashing algorithm PBKDF2. A desired key size of 512 bits is specified in terms of 32-bit words. The method ‘users(email)’ is called from the smart contract instance to confirm if the user email already exists.

If the email does not already exist, then the new user account details (email, salt, and hashed password) are sent to the smart contract method ‘CreateAccount’ for handling, with the ‘from’ field specifying the Ethereum account address. Once the password has been salted, hashed by PBKDF2, and sent to the contract to be handled, the user is sent to the login page, indicating successful account creation. For testing purposes, the user’s email, salt, and password are stored in local storage for viewing. These user credentials entered are dummy credentials. Logs of timestamps, transaction hashes, and block numbers have been added for testing. Listing 8 shows the implementation of a contract call.

**Listing 7.** JavaScript implementation of password salting and hashing along with existing user check.

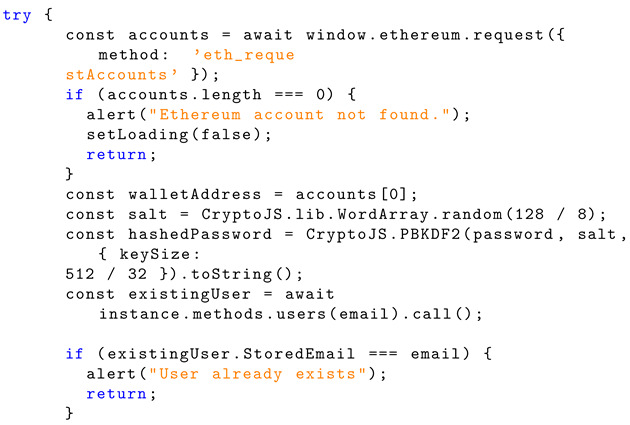



**Listing 8.** JavaScript implementation of a contract call.

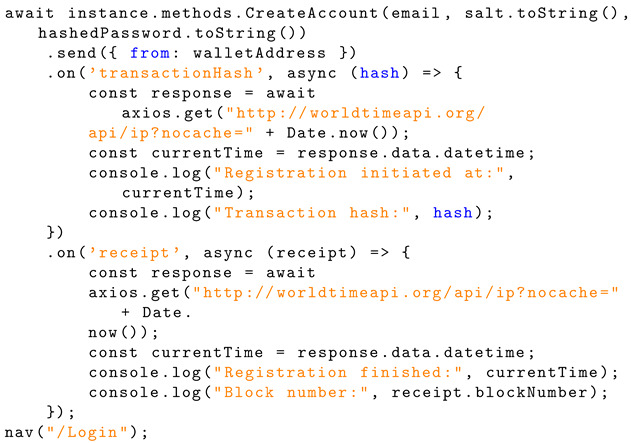



### 4.4. Login

The login page also uses the libraries ‘useState’, ‘useEffect’, ‘loadWeb3’, ‘loadBlockchainData’, and ‘useNavigate’. The state variables within the functional component are declared with the hook ‘useState’ in the same fashion as the register page, along with ‘useNavigate’. The blockchain data are also loaded using the function ‘userAccounts’ as with the registration page. The smart contract instance is called again to find the corresponding salt for the user in ‘StoredSalt’. Then, it calculates the hash for the password entered by the user in the same fashion as when creating an account but using the salt that is stored. The library CryptoJS is again used for this purpose. Listing 9 shows the javascript implementation for fetching users and salt generation.

**Listing 9.** JavaScript implementation: fetching users and salt generation using ‘StoredSalt’.

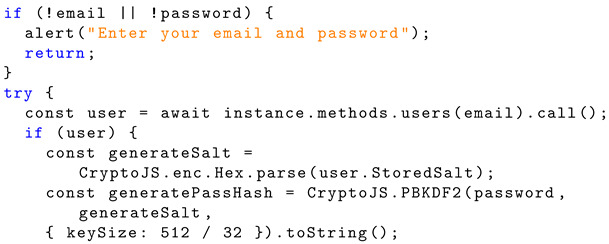



If the generated hashed password and the wallet address match what is stored for the account associated with the email, then the user can navigate to their home page. The email and account info will be saved to local storage to be called at another point. The JavaScript code below sends the successful login attempt to the ‘LoginAttempt’ function within the smart contract instance. Listing 10 shows the Solidity structure schema for the account finder.

The ‘LoginAttempt’ function emits an event when the user successfully logs into their account. Listing 11 shows the implementation of successful login to the blockchain.

**Listing 10.** JavaScript implementation of account finder.

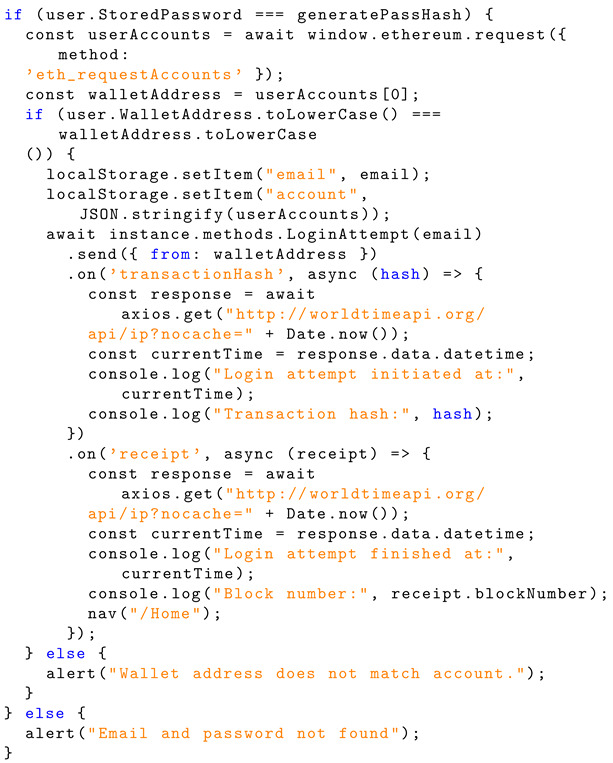



**Listing 11.** Solidity implementation of login to the blockchain.

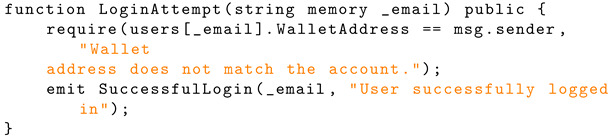



### 4.5. Home

The ‘useEffect’ hook is utilised to interact with the blockchain and update the ‘setInstance’ with the contract instance. Listing 12 shows the JavScript implementation of the ‘useEffect’ hook.

**Listing 12.** JavaScript implementation of ‘useEffect’ hook.

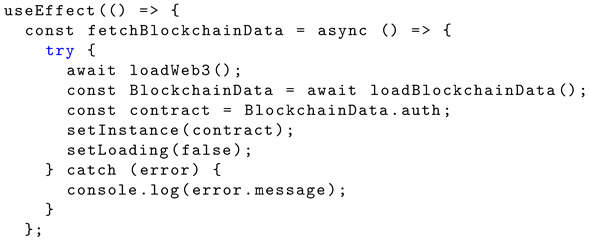



A basic function handling account logout allows the user to navigate back to the login page. Listing 13 shows the JavaScript implementation of logout functionality.

**Listing 13.** JavaScript implementation of logout confirmation.

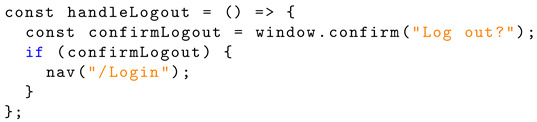



The user email is sent to the ‘DeleteAccount’ function within the smart contract instance to handle deletion. Listing 14 shows the JavaScript implementation of sending information to the smart contract for handling.

The ‘DeleteAccount’ function uses the delete keyword to reset the state of a data structure or variable. This is used to delete a user account. This event is emitted to the blockchain. Listing 15 shows the implementation information for account deletion.

To ensure that the user follows the password requirements, the ‘passRequirements’ function is called in the same fashion as shown for the registration page. Then, a new salt is generated, and the new password is hashed using PBKDF2 again. The email, new salt, and new hashed password are sent to the ‘UpdatePassword’ function within the smart contract instance to handle this. Listing 16 shows the implementation information for information exchange.

The ‘UpdatePassword’ function overwrites the ‘StoredSalt’ and ‘StoredPassword’ for the account and emits this to the blockchain. Listing 17 shows the Solidity implementation for password change.

**Listing 14.** JavaScript implementation of sending information to a smart contract for handling.

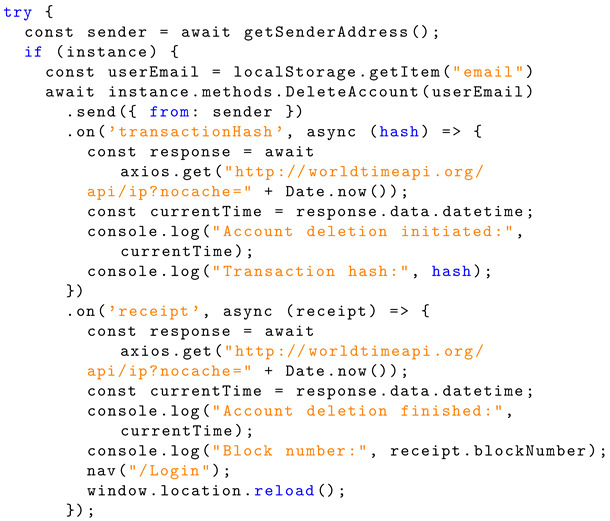



**Listing 15.** Solidity implementation of account deletion and emission to the blockchain.

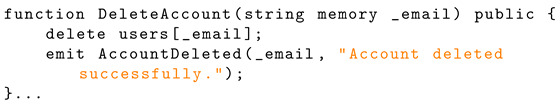



**Listing 16.** JavaScript implementation of sending password-related information to a smart contract for handling.

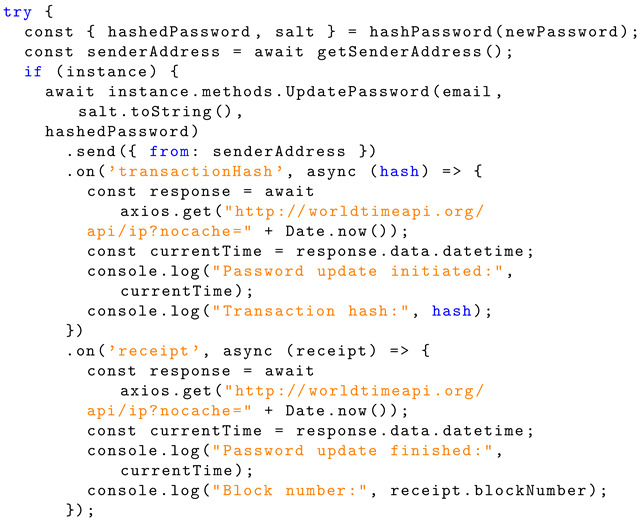



**Listing 17.** Solidity implementation of password change.

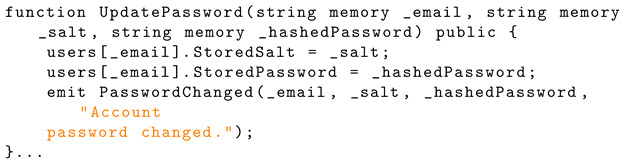



### 4.6. Truffle Config

The Truffle config file must be configured to allow a connection to both Ganache and the Sepolia network. The library ‘HDWalletProvider’ is required to connect to Sepolia. The MetaMask mnemonic phrase and the Infura project ID were specified. The estimated gas was 616,909 when the smart contract was deployed on the Sepolia network.

## 5. Results and Discussion

The proposed authentication framework, which includes two smart contracts, was deployed to the Sepolia network. In Table 1, the transaction hashes, block number, contract address, gas used, and gas price can be viewed, along with other information such as the address and balance of the MetaMask wallet. The two contracts cost ETH 0.001181576819284785 in total for deployment.

Once the contracts were deployed, the application was started to create transactions of user interaction to the blockchain. Transactions for account creation, login, password change, and account deletion were done multiple times during different times of the day to find an average cost for each interaction on the Sepolia network. These transactions can be viewed within Etherscan to discover gas used, gas price, and transaction fees, along with emitted events. The email address of the user, their salt, hashed password, wallet address, and description of the event were emitted to the blockchain. This was done for testing purposes and to confirm that the salting and hashing were being performed successfully. However, it would not be done in a real environment. The salt especially should not be viewable to the public, as this could allow someone to reverse-engineer the hash to discover the plain text. The description emitted allowed for easy identification of each transaction type, e.g., account creation. The transactions can be seen in Etherscan, as shown in Figure 2.

### 5.1. Transaction Throughput

Developing cost-effective applications while also finding the balance between usability is a challenge many DApp developers face. Services are available to provide estimates of how long transactions will take to be processed at a certain gas price. The higher the gas price, the more likely it will be processed faster. Research by Pacheco et al. found that transactions on the Ethereum network were processed in a median of 57 s. They also found that 90% of the transactions were processed within 8 minutes. Their research found that, while higher gas prices did result in faster processing, there was no obvious difference between expensive and very expensive transactions [43].

As shown below, ‘Low’ prices could take 10 min to process, whereas ‘Average’ and ‘High’ prices were much faster at 3 min. At the time of deployment, gas prices were checked in the Etherscan gas tracker, which comes out to be 35 gwei, 36 gwei, and 36 gwei for low, average, and high cases, respectively.

Research by [44] indicated that the average throughput for Ethereum was estimated at 4 transactions per second at 8,100,000 blocks. At 4,900,000 blocks, the throughput was 16.7 transactions per second. This is a minuscule number in comparison to the figures presented by VISA and PayPal. This implies that Ethereum is not ready to support real-time Internet applications.

### 5.2. Transaction Fees

Transactions on Ethereum require a transaction fee, which is calculated by gas usage multiplied by gas price. Gas usage is the amount of computational power required to process it. The base fee is the lowest amount the network requires at the time of the block. The max Priority fees are the maximum amount the user is willing to spend. The gas price is the amount of Ether the sender paid per unit of gas consumed in the processing. This allows a sender to incentivise the processing of their transaction [43].

The gas usage for each transaction type is shown below. The transaction with the highest gas usage is account creation. The password change transaction was the second highest. This is due to brand-new salts and hashes being generated in these transactions. More gas is needed due to the increased computational power being utilised. Table 2 shows the gas usage for a transaction involving a user transaction.

In the Table 3, an average was calculated for all samples taken for transaction fees and gas prices. It should be noted that Sepolia is not the main network of Ethereum, and gas prices are significantly lower.

Transaction details for login transactions can be seen in Table 4.

Table 5 shows the transaction details on the blockchain for password change. It shows the transaction fee that needs to be paid for a transaction involving change of a password by a user. This table also shows the gas price (gwei) at the time of a transaction for a change in password. For analysis purposes, values of these two parameters, i.e., transaction fee and gas prices, are shown at the block level and also for a few particular blocks identified with block numbers.

Table 6 shows the transaction details on the blockchain for account deletion. It shows the transaction fee that needs to be paid for a transaction involving deletion of a user account. This table also shows the gas price (gwei) at the time of a transaction involving user account deletion. These values are given for a block as a whole and also for a few blocks so that account holders can analyse the associated cost.

Deploying to the Sepolia Test Net provided the opportunity to ascertain the gas usage of the contract’s deployment and transactions. As the gas usage is the same even when deploying to the Ethereum Main Net, it is now possible to compare what the price would be if deployed to it. The Etherscan Gas Tracker tool (accessible at www.etherscan.io/gastracker) revealed that the average gas price for the Main Net from 18 March 2024 to 22 March 2024 was around 33.4 gwei. Figure 3 shows the average gas prices on different dates, from 18 March 2024 to 22 March 2024.

The heat map provided by the Etherscan Gas Tracker tool also showed what times had the highest gas prices during this week. Figure 4 shows the Etherscan heatmap.

Before 9 a.m., transactions were cheaper. From 9 a.m. onward, it saw a massive increase, with the highest being 96.81 gwei on Tuesday, 19 March, at 9 a.m. Prices tended to become cheaper later into the night and early morning. Multiplying the gas usage of each transaction for the five samples above gives an estimate of how much it costs to confirm the transactions. This can be converted into Ether and, finally, the equivalent GBP price can be calculated. The GBP prices were calculated based on the exchange rate on 30 March 2024. Table 7 shows the gas estimate costs for migrations, authentication, account creation, user login, password change, and account deletion.

## 6. Performance Evaluation

### 6.1. Response Time and Latency

To calculate the time taken for each transaction to be added to the blockchain from its initiation, a log was implemented using a timestamp. The following code, as shown in Listing 18, was added to the application to achieve this functionality.

**Listing 18.** Timestamp implementation in JavaScript.

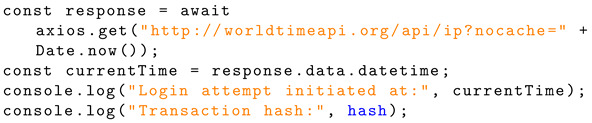



A popular API ‘www.worldtimeapi.org’ (accessed on 13 May 2024) was used to log the timestamps to ensure the time logged was as accurate as possible. Other functions, such as ‘Date toISOString()’, can be used to return a string with the date and time based on the host PC time. For this reason, it is essential that the time on the host PC is as accurate as possible to be able to determine the time differences of the transactions down to the second. However, despite our best efforts, the host PC used in the testing could be up to 0.3 s ahead or behind. This was mitigated by logging timestamps using worldtimeapi.org instead. This input is logged to the console for easy viewing. The initial execution of the transaction through MetaMask (the moment the user clicks the accept button on MetaMask to authorise the transaction) was noted, along with the final confirmation to the blockchain (viewed on Etherscan). Multiple samples were taken at busy and quiet Sepolia network times to yield an average. Table 8 shows the time averages for account creation, user login, password change, and account deletion.

To determine the time delay between the moment the user clicks the button to call MetaMask (e.g., create an account/login button) to the time in which the application receives verification from MetaMask that the transaction was confirmed successfully, a console log was added. When the user clicks the login button after inputting their credentials, assuming they are correct, MetaMask is called and asks the user to confirm the transaction. As this time is dependent on how quickly the user can authorise the transaction, this will vary. Multiple samples were taken at busy and quiet network times to find an average of how long it takes from the user’s initial interaction with the application to the application recognising the confirmed transaction. The results gathered for the initiation/confirmation time and total time varied depending on how busy the network was at the time. A larger sample result would have shown that the highest gas usage transaction would likely have had the longest confirmation time and total time. However, due to the small sample size, this is not reflected in these results.

### 6.2. Comparative Analysis

Multiple login attempts were conducted on the Outlook application to estimate the delay between when the user clicked the login button and the successful sign-in to Outlook. This was conducted on an account with 2FA enabled and measured the time between the login, authenticating the OTP, and the arrival on the home page. The results from the application were compared to the Outlook results, and an average was calculated. Only the speed of the login page was calculated as registering, as password change and deletion of an Outlook account have multiple steps that take time to finish. The application does not have these steps, and a comparison of both applications would not yield an accurate result. Table 9 shows the time averages for the Outlook use case.

The results of the comparison of the application and Outlook are shown in Table 10.

While on the Sepolia network, the authentication process was faster for the application than Outlook. However, this would not be the case on the main Ethereum network, as confirmation time can take up to 3 min for the ‘Average’ price, as discussed previously. While ‘High’ fees are typically done within 30 s, this is still a long time for the user to wait to access their account. At extremely busy times on the Ethereum network, it can take up to 3 min, even for ‘High’ fees.

The current code waits for the transaction to be confirmed to the blockchain before allowing the user access to their account. To remedy this, the code could be changed slightly to allow the user to log into their account as soon as the transaction has been initiated in MetaMask, assuming their account creation has already been confirmed to the blockchain. However, the implications of allowing users access to the account before the transaction has been confirmed to the blockchain have not been explored in this research.

For account creation, the user will not be able to log in until after the transaction has been confirmed to the blockchain. As discussed previously, this could take a long time. An off-chain solution such as a database could be implemented to store user data for authentication while the account creation transaction is being confirmed to the blockchain. This would allow the user to log in straight away while the transaction is pending but defeats the purpose of this research, as they will not be able to use MetaMask to authenticate this login.

The time needed for the user to log in to their account could be much faster than the method used by Outlook, depending on how busy the blockchain network is at the time. It is also dependent on how quickly the user can press the login button and subsequent confirm button on their MetaMask wallet.

### 6.3. Scalability

Scalability refers to the ability to manage an increased workload. This can be done without the necessity of additional resources or by implementing cost-effective strategies to expand a system’s capacity in response to increased demands. As more users sign up for a service, an efficient method of scalability is required. An efficient scalable system should exhibit improved performance as the system increases. The increase in performance should align directly with the additional number of units added, namely, in this case, users [45].

For this research, scalability becomes a concern for several reasons. The ability to test the scalability of this application relies on creating potentially millions of user accounts. However, this is extremely resource-intensive and requires substantial computational power, bandwidth, storage, and time to execute. This is impractical and cost-prohibitive for a single person or even a small team.

To host millions of users, infrastructure that includes servers and network architecture is required for effective scalability. The cost and time of maintaining this infrastructure exceeds what is possible for individual researchers. The creation of millions of accounts is a time-consuming task and would not line up with the timeline of this project. Multiple people may need to be added to the project to scale up to the number of users required to finish in time. On top of this, the application may require monitoring and troubleshooting, which are also time-consuming. Meanwhile, managing the data retrieved from this research would also become complex and require a substantial amount of time to compile and analyse. For these reasons, the testing of the scalability of this application is not possible in the current study.

## 7. Conclusions and Future Recommendations

### 7.1. Conclusions

In conclusion, the implementation of blockchain technology as a form of 2FA presents promising advantages, such as increased security and convenience, in comparison to methods such as OTPs, biometrics, hardware-based tokens, and authentication through databases. However, the process of confirming a transaction to the blockchain can be slow and may necessitate the integration of databases to store and retrieve data to allow users to log into their account without interruption after account creation. As discussed in the previous section, this defeats the purpose of this research, as there is no way to authenticate this login to the blockchain until after the account creation transaction has been confirmed. User interactions that require confirmation to progress could potentially have users waiting for a substantial amount of time. This could irritate users. Likewise, deploying a smart contract and the subsequent transactions can be pricey and unpredictable due to the nature of cryptocurrency and its constantly changing market values. Peak times for the Ethereum network appear to be 9 a.m. onward, and prices can soar. An organisation with a limited budget may find it not worth investing in.

While this research has highlighted potential benefits of blockchain technology, there are unexplored implications that warrant further investigation, such as scalability and integration of off-chain storage. Also, this research did not discuss the feasibility or implications of on-chain storage in detail.

Further research is required to delve into the practical implementation of blockchain 2FA methods, such as regulatory compliance of user data, user experience (no real users were asked to use the application and give feedback), and the likeliness that organisations would adopt this method. By addressing these considerations, a more comprehensive understanding of the feasibility of this method can be considered. As blockchain technology continues to advance, the potential for offering enhanced authentication solutions is promising in an increasingly hostile digital world.

### 7.2. Future Recommendations

Based on the findings of this research on the implementation of the blockchain as a form of 2FA, the following recommendations have been proposed for future research and practical applications of this technology.

Further Research: Additional research should be conducted to further explore the feasibility and implications of using blockchain technology as a 2FA mechanism. This could include potential vulnerabilities, compatibility with existing software, scalability, and user acceptance.

Development: Develop a prototype web application deployed on the Ethereum Main Net to test the viability of the authentication method in a real-world environment. This could include implementing and testing cryptocurrency wallets other than MetaMask to give users a choice of which wallet they wish to use.

Integration: Explore integration with other platforms, such as databases and decentralised data storage networks, for the adoption of blockchain-based 2FA. Compatibility with devices such as tablets and mobile phones should be carefully considered.

User Testing: Once the prototype has been deployed, allow users to test the application and gather feedback to address usability issues and concerns in the early stages.

Security: Explore potential vulnerabilities and compliance considerations while working with user data. Implement security measures to protect user data and ensure legal compliance.

Continuous Advancements: Continue to improve and innovate the application based on feedback and emerging technology. Stay on top of advancements in blockchain technology and cyber security developments.

These recommendations may encourage future researchers to explore the potential of blockchain technology, especially in the space of cyber security, and contribute to the development of secure authentication methods.

## Figures and Tables

**Figure 1 sensors-24-05830-f001:**
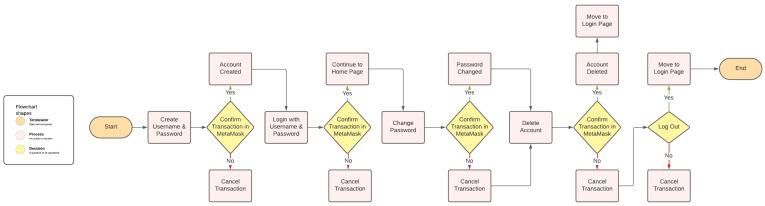
The flowchart of the proposed authentication mechanism.

**Figure 2 sensors-24-05830-f002:**
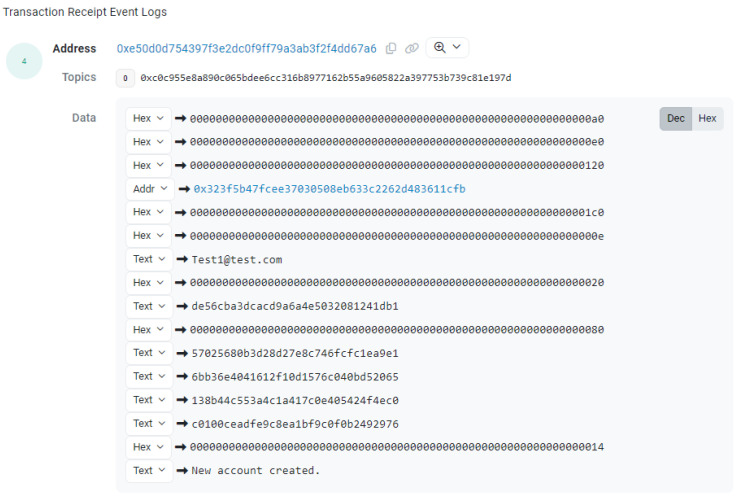
Transaction viewed in Etherscan.

**Figure 3 sensors-24-05830-f003:**
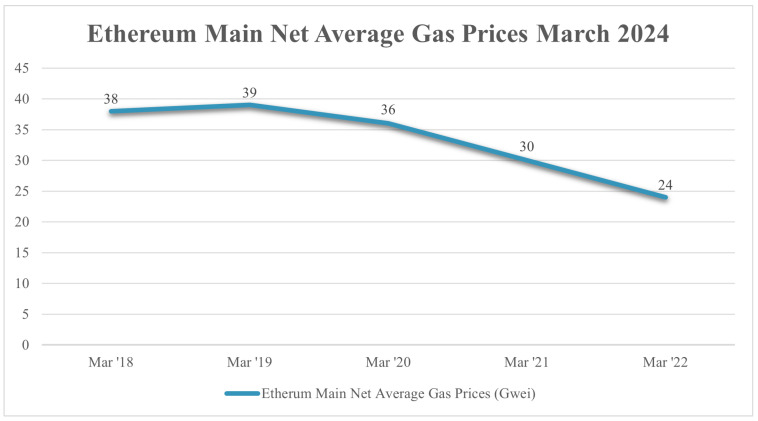
Average gas prices in gwei.

**Figure 4 sensors-24-05830-f004:**
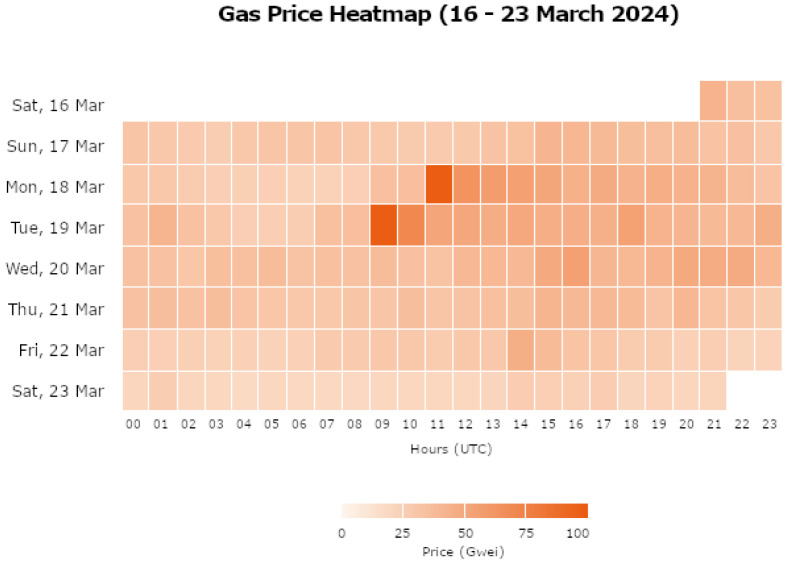
Etherscan heatmap.

**Table 1 sensors-24-05830-t001:** Transaction information for the deployment of Migrations.Sol.

Attribute	Value	
Transaction Hash	0x351180a538aef925a3242f2b23e43715 db72b693692cf9601b85390d8c857720	0x0a7b43a4d7d634848e6d07b1ec1865 cdd7dbc555ce723eda09b2272a5b651400
Blocks: 1	Seconds: 4	Seconds: 8
Contract Address	0xF557975252F073f7b12D 451C4043607693590374	0xE50D0d754397F3E2D C0F9Ff79a3AB3f2F4dD67A6
Block Number	5567940	5567942
Block Timestamp	1711497828	1711497852
Account	0x323f5B47Fcee3703050 8eb633c2262D483611CFB	0x323f5B47Fcee37030508 eb633c2262D483611CFB
Balance	0.2237	0.2226
Gas Used	176,751 (0x2b26f)	704,514 (0xac002)
Gas Price	1.340773569 gwei	1.340773569 gwei
Value Sent	ETH 0	ETH 0
Total Cost	ETH 0.000236983069094319	ETH 0.000944593750190466

**Table 2 sensors-24-05830-t002:** Gas usage for user interaction transactions.

Attribute	Value
Create Account	233,395
Login	27,438
Change Password	63,345
Delete Account	57,140

**Table 3 sensors-24-05830-t003:** Transaction details for account creation.

Attribute/Value	Mar-27	Mar-27	Mar-27	Mar-27	Mar-29	Average
Block	5,567,990	5,572,493	5,573,555	5,574,098	5,586,357	–
Transaction Fee (ETH)	0.000588420490974580	0.001415432873583820	0.000611064729675500	0.000604978046443765	0.000704553131663110	0.0007848898544681550
Gas Price (Gwei)	2.521135804	6.064538116	2.6181569	2.592078007	3.018715618	3.36292488900

**Table 4 sensors-24-05830-t004:** Transaction details for login.

Attribute/Value	Block	5,568,001	5,572,511	5,573,563	5,574,102	5,586,365
Transaction Fee (ETH)	0.000068973420094350	0.000148450463721708	0.000071700416240010	0.000071826204763242	0.000084542480364318	0.0000890985970367256
Gas Price (Gwei)	2.513791825	5.410396666	2.613179395	2.617763859	3.081218761	3.24727010120

**Table 5 sensors-24-05830-t005:** Transaction details for password change.

Attribute/Value	Block	5,568,005	5,572,519	5,573,574	5,574,107	5,586,370
Transaction Fee (ETH)	0.000159070924128480	0.000324438412013325	0.000173590251595500	0.000165740495424105	0.000197767302759225	0.0002041214771841270
Gas Price (Gwei)	2.511183584	5.121768285	2.7403939	2.616473209	3.122066505	3.22237709660

**Table 6 sensors-24-05830-t006:** Transaction details for account deletion.

Attribute/Value	Block	5,568,007	5,572,525	5,573,580	5,574,112	5,586,374
Transaction Fee (ETH)	0.000143433466710920	0.000285686271543600	0.000165717073217740	0.000147904620627760	0.000185380323034660	0.0001856243510269360
Gas Price (Gwei)	2.510211178	4.99975974	2.900193791	2.588460284	3.244317869	3.24858857240

**Table 7 sensors-24-05830-t007:** Gas estimates for migrations, authentication, account creation, user login, password change, and account deletion.

Gas Price (Gwei)	38	39	36	30	24	Average
			Migrations			
Gas Used	176,751	176,751	176,751	176,751	176,751	–
Transaction Fee (Gwei)	6,716,538.00	6,893,289.00	6,363,036.00	5,302,530.00	4,242,024.00	5,903,483.40
Transaction Fee (ETH)	0.006716538	0.006893289	0.006363036	0.00530253	0.004242024	0.005903483
GBP	18.70	19.19	17.71	14.76	11.81	16.44
			Authentication			
Gas Used	704,514	704,514	704,514	704,514	704,514	–
Transaction Fee (Gwei)	26,771,532.00	27,476,046.00	25,362,504.00	21,135,420.00	16,908,336.00	23,530,767.60
Transaction Fee (ETH)	0.026771532	0.027476046	0.025362504	0.02113542	0.016908336	0.023530768
GBP	74.53	76.49	70.61	58.84	47.07	65.51
			Account Creation			
Gas Used	233,395	233,395	233,395	233,395	233,395	–
Transaction Fee (Gwei)	8,869,010.00	9,102,405.00	8,402,220.00	7,001,850.00	5,601,480.00	7,795,393.00
Transaction Fee (ETH)	0.008869010	0.009102405	0.0084022200	0.0070018500	0.0056014800	0.0077953930
GBP	24.69	25.34	23.39	19.49	15.59	21.70
			User Login			
Gas Used	27,438	27,438	27,438	27,438	27,438	–
Transaction Fee (Gwei)	27,400.00	27,399.00	27,402.00	27,408.00	27,414.00	27,404.60
Transaction Fee (ETH)	0.000027400	0.000027399	0.000027402	0.000027408	0.000027414	0.000027405
GBP	0.076282	0.076279	0.076288	0.076304	0.076321	0.076295
			Password Change			
Gas Used	63,345	63,345	63,345	63,345	63,345	–
Transaction Fee (Gwei)	63,307.00	63,306.00	63,309.00	63,315.00	63,321.00	63,311.60
Transaction Fee (ETH)	0.000063307	0.000063306	0.000063309	0.000063315	0.000063321	0.000063312
GBP	0.176248	0.176245	0.176254	0.176270	0.176287	0.176261
			Account Deletion			
Gas Used	57,140	57,140	57,140	57,140	57,140	
Transaction Fee (Gwei)	57,102.00	57,101.00	57,104.00	57,110.00	57,116.00	57,106.60
Transaction Fee (ETH)	0.000057102	0.000057101	0.0000571040	0.0000571100	0.0000571160	0.0000571066
GBP	0.158973	0.158970	0.158979	0.158995	0.159012	0.158986

**Table 8 sensors-24-05830-t008:** Time averages for account creation, user login, password change, and account deletion.

Attribute/Value	Mar-27	Mar-27	Mar-27	Mar-27	Mar-29	Average
			Account Creation			
Initiation from MetaMask	0:14:09	16:07:23	19:52:26	21:47:13	17:26:49	–
Confirmation Time	0:14:36	16:07:36	19:52:36	21:47:24	17:27:12	–
Total Time (Seconds)	0:00:27	0:00:13	0:00:10	0:00:11	0:00:23	0:00:17
			User Login			
Initiation from MetaMask	0:16:54	16:10:59	19:54:22	21:48:23	17:28:51	–
Confirmation Time	0:17:00	16:11:12	19:54:36	21:48:24	17:29:00	–
Total Time (Seconds)	0:00:06	0:00:13	0:00:14	0:00:01	0:00:09	0:00:09
			Password Change			
Initiation from MetaMask	0:17:39	16:12:41	19:55:50	21:49:13	17:29:50	–
Confirmation Time	0:17:48	16:12:48	19:56:48	21:49:24	17:30:12	–
Total Time (Seconds)	0:00:09	0:00:07	0:00:58	0:00:11	0:00:22	0:00:21
			Account Deletion			
Initiation from MetaMask	0:18:05	16:13:53	19:57:55	21:50:02	17:31:07	–
Confirmation Time	0:18:12	16:14:00	19:58:00	21:50:24	17:31:12	–
Total Time (Seconds)	0:00:07	0:00:07	0:00:05	0:00:22	0:00:05	0:00:09

**Table 9 sensors-24-05830-t009:** Individual Function’s Time average for Outlook.

Attribute/Value	Mar-23	Mar-24	Mar-24	Mar-25	Mar-26	Average
Button Clicked	23:55:45	16:00:15	23:45:49	15:31:40	20:14:39	–
Landed on Homepage	23:56:03	16:00:31	23:46:13	15:31:57	20:14:54	–
Total Time	0:00:18	0:00:16	0:00:24	0:00:17	0:00:15	0:00:18

**Table 10 sensors-24-05830-t010:** Overall Time average for Outlook.

Attribute/Value	Attempt 1	Attempt 2	Attempt 3	Attempt 4	Attempt 5	Average
Total Time App	0:00:13	0:00:19	0:00:22	0:00:07	0:00:15	0:00:15
Total Time Outlook	0:00:18	0:00:16	0:00:24	0:00:17	0:00:15	0:00:18

## Data Availability

Data are contained within the article.

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
