# Peer review of "A Blockchain-Based Authentication Mechanism for Enhanced Security"

_sensors, 2024, doi:10.3390/s24175830_

Round 1

Reviewer 1 Report

Comments and Suggestions for Authors

This manuscript fascinatingly explores the use of blockchain technology—especially Ethereum—for two-factor authentication (2FA).  This strategy has great promise for improving user authentication process security.  The work demonstrates a thorough understanding of blockchain technology and the challenges associated with verification.

Strengths: The topic is highly relevant to cybersecurity concerns.  The implementation details are well-explained, providing a clear picture of the proposed system.  Comparing it with traditional 2FA methods offers valuable insights.  Discussing potential challenges and limitations shows critical thinking.

But there are some areas for improvement.  The sample size for performance testing is relatively small.  Expanding this sample size would yield more robust results.  The current analysis of scalability is limited.  Given its importance for real-world applications, a more in-depth exploration of this aspect would strengthen the paper.  The practical implications of implementing this system, including user experience and adoption challenges, could be discussed more thoroughly.  The comparative analysis with existing 2FA methods could be more extensive, including a broader range of current solutions.  Expanding on the security analysis of the proposed system would be beneficial, particularly in addressing potential vulnerabilities specific to blockchain-based authentication.

The suggested improvement to this manuscript is to conduct more extensive performance testing, particularly in an Ethereum manner or a more representative test environment.  Your proposed system should include a more detailed discussion of the security, usability, and cost trade-offs.  Explore the regulatory and compliance implications of storing authentication data on a public blockchain.  Conduct user testing to gather feedback on the usability of the proposed system.  Strengthen the conclusion by more clearly articulating this work's unique contributions and potential impact on the field.  Your research shows promise in advancing authentication methods using blockchain technology.

With some refinement and expansion in key areas, this work could significantly contribute to cybersecurity and blockchain applications.  This is a solid foundation for an exciting and potentially impactful study.  I encourage the author to build upon this work, addressing the suggested improvements to enhance its scientific rigor and practical relevance.

Reviewer 2 Report

Comments and Suggestions for Authors

The paper titled "The Blockchain-based Authentication Mechanism for Enhanced Securit" implement the Authenifation based on the Blockchain.  I think that, Authors has not proposed any new authentication protocol they has only developped a web based application that implement the authentications protocol.

Author should proposed a new protocol and details its different phases including (registration phase, login and authentication phase, and password change phase)

A) The abstract should be rewritten to highlight the main purpose of this work and present shortly the obtained results

B) I didn't find the Figure 3.7  (my be the text is copied from an other paper or web page)

C) The introduction is not well writen, Author have to rewrite itroducing the issue and some ackground the importance of the authentication, then the various of authentication solutions

D)It should be clearly demonstrated that this study is new, useful and different(in the end of introduction)

E) The Section two give background information about Blockchaine, Authentication, Password.. (So, I suggest to rename this section to Background)... Furthermore, Reated work section shoud contains some previous works that deal with the some work that author discusses in this paper.

F) Why author   Explain the quantitative research in the first paragraph of  section 3.  (this paragraph don't contain any important information related to this section. So, I propose to delete it )

G) What you mean by those two sentences :

Table 5 shows the transaction details of password change.

Table 6 shows the transaction details for account deletion.

Round 2

Reviewer 2 Report

Comments and Suggestions for Authors

Author has done required change